# Elite athletes' lifestyles: Consumerism to professionalism

**Ehsan Mohamadi Turkmani** [1] *, **Hamid Reza Safari Jafarloo**[2], **Amin Dehghan Ghahfarokhi**[1]

**1** Department of Sport Management, Faculty of Physical Education and Sport Sciences, University of Tehran, Tehran, Iran, **2** Jahrom University, Jahrom, Iran

* ehsan.mohamadi@ut.ac.ir

**Data Availability Statement:** Data (excerpts from interview transcripts) are provided as supporting information.

## Abstract

Today, elite athletes form an important social group, and the non-sport facets of their lives matter as much as their sports performances. However, there has been little empirical research on the lifestyles of elite athletes. Therefore, this study aimed to develop knowledge about the Iranian elite athletes' lifestyle. The study was conducted with a qualitative approach in two phases. Glaserian grounded theory was used in the first phase, and thematic analysis was used in the second phase. Participants of the first phase included 19 sports experts, such as sports sociologists, sports psychologists, and sports coaches, who were selected by purposive and snowball sampling methods for holding unstructured in-depth interviews. The data were simultaneously analyzed using a set of open, theoretical, selective coding and memos. The codes were grouped into three different categories with different natures. The emerged theory advanced our understanding of the lifestyle shaping structures of elite athletes, lifestyle indicators, and even professionalization of their lifestyles. According to the results, the Iranian elite athletes' lifestyles include indicators, such as professional mindset, competencies, life vision, financial literacy, responsibility, consumption, leisure, personal issues, and religious behavior. Subjects of the second phase were 44 Iranian athletes in the national levels who participated voluntarily in the study. The data were analyzed by thematic analysis method, and lifestyles typologies were identified. Based on results, five dominant lifestyles among the Iranian elite athletes were identified: consumerist, easy going, socially useful, profit-oriented, and professional. Finally, the features of each lifestyle were discussed.

## Introduction

It seems that being a professional athlete could be considered a profession and a lifestyle because success at the highest level of professional sports is almost impossible without devoting many hours of life [1]. Horley, Carroll, and Little (1988) believed that lifestyle is a common term in everyday usage [2]. Scholars in various fields have offered various definitions of lifestyle. Giddens (1991) said that lifestyle is a relatively harmonious set of all the behaviors and activities of a certain person in daily life, requiring a set of habits and orientations and,

**Funding:** The author(s) received no specific funding for this work.

**Competing interests:** The authors have declared that no competing interests exist.

therefore, having a kind of unity. Often in social sciences, lifestyle is used as a concept in discussing social inequalities, and new forms of social differentiation in postmodern contexts are described by this concept [3–5]. The concept of lifestyle was first introduced in 1929 by the social psychologist Alfred Adler. According to him, lifestyle is the central character with an individual interpretation, not a social one. Adler believed that lifestyle is mainly established in the humans' early years of life up to about five years. He stated that "... Lifestyle denotes a person's basic character established early in childhood governing his reactions and behavior". He considered lifestyle as creativity, which results from coping with limitations, obstacles, contradictions, and crises that a person expresses in the path of progress towards his aims [6]. As another theorist in this field, Weber considered being selective as the main feature of lifestyle. However, the choices are made in a range of structural constraints and bottlenecks, limiting the possibility of making choices. In Weber's theory, "life chance" and "life conduct" are seen as two distinct and at the same time interdependent dimensions of lifestyle. None of them can be understood without the other [3]. Simmel is another theorist who raised the issue of fashion about lifestyle. In his book entitled "Fashion," he explored the reasons for the multiplicity of fashion changes in modern culture. He believed that clothes were designed to show the social status of people. Fashion can be described as a way of life revolving around activities and interests and is often shaped by cultural values influencing gender, age, and social attitudes [7]. Indeed, fashion has become an essential part of the human experience and is related to people's aesthetic preferences [8]. Regarding the lifestyle, Veblen (1899) described the phenomenon of leisure and lifestyle of the leisure class in his book entitled "The Theory of the Leisure Class." Veblen used the term "conspicuous consumption" to describe leisure class more accurately. Conspicuous consumption refers to the use of things regardless of their expenditure. This form of consumption brings credit to the individual [9]. In this regard, Law, Bloyce, and Waddington (2020) researched the consumption behavior of famous English football players based on this theory. Their research findings showed an interesting topic on Veblen's theory of conspicuous consumption about consumption behavior of footballers. Veblen specifically pointed to the importance of clothing as an aspect of conspicuous consumption, noting that people can display wealth through purchasing many properties. However, the cost of clothing as a method has a fundamental advantage over other methods. "Our apparel is always in evidence and affords an indication of our pecuniary standing to all observers at first glance." Law, Bloyce, and Waddington (2020) stated that professional footballers display a special image of a professional footballer, making even lower-income players buy expensive clothes and accessories to be accepted by the others. Players who do not match the expected image may be subjected to sanctions by their teammates [10].

In Bourdieu's theory, lifestyle includes categorized actions of the individual in areas, such as division of day and night, type of leisure and sports, ways of socializing, furniture and home, and the etiquette of speaking and walking. The lifestyle is embodied by the preferences of individuals [11]. In his book entitled "Distinction," Bourdieu placed a strong emphasis on the effect of class hierarchy on behavior and considered class as the most decisive variable in determining lifestyle [12]. The most central concept in Bourdieu's thought is the concept of ″Habitus″, forming the core of his explanation for human actions. According to Bourdieu (1990), Habitus is defined as "systems of durable, transposable dispositions, structured structures predisposed to operate as structuring structures, that is, principles generating and organizing practices and representations that can be objectively adapted to their outcomes without presupposing a conscious aiming at ends or expressing mastery of the operations necessary to attain them" [13]. Bourdieu (1978) considered the social class of people to be related to their physical strength and the way they protect and nourish their body and stated that it refers to the differentiation of sports at these social classes according to different social degrees.

Generally, individual and fantasy sports belonged to the upper classes of society, group sports belonged to the middle classes, and individual sports with an emphasis on physical strength belonged to the lower classes [14]. Some sports sociologists have used this theory to study consumption behavior in sports. Based on Bourdieu's theory, Kahma (2012) investigated lifestyle choices in Finland's sports fields, focusing on differences in occupational class and other contextual variables. The results showed a difference between job levels in sports participation and involvement in individual sports' activities. However, these differences are less pronounced in sports' spectators [15]. Gemer (2020) also performed his study of consumer lifestyle in Canadian professional sports based on Bourdieu's theory. He concluded that cultural omnivores are the most important consumer group in Canadian professional leagues. The cultural 'omnivore' refers to ". . . A cultural consumption profile traversing the line between traditionally highbrow and lowbrow cultural forms" [16]. According to Bourdieu's theory of distinction (one of the valuable lifestyle-related theories), there are differences between people's lifestyles with different hobbies and sports. Bourdieu (1984) pointed out that there are distinctive lifestyles in a society like France whose favorite hobbies are water skiing, tennis, horseback riding, chess, golf, and playing or listening to the piano. On the opposite spectrum, we are dealing with football, fishing, and playing or listening to other types of music.

Although many studies have been done in lifestyle and sports, these studies have often been performed on non-athletes who have considered sports as one of their distinguishing indicators, while in the lifestyle of athletes; this type of research is minimal. On the other hand, the basis for distinction can be considered as differences of individuals in some indicators that are very diverse from the point of view of theorists, regarding diet, self-decoration, type of housing, type of means of transportation, and leisure to artistic, social, and sports activities (see Table 1). These studies have only been done on ordinary people (not athletes); herein, the general theories of lifestyles are usual, and sports activities are a lifestyle indicator that could help people distinguish themselves from others.

Reflecting on the table presented above, some questions may arise. For example: How can the indicators mentioned by theorists be the basis for the distinct lifestyle of elite athletes? Or which one of them can be considered in the lifestyle of athletes?

It is believed that elite athletes have special living conditions that necessitate theorizing about their lifestyle. In the new era, professional sports have become a high-traffic industry.

**Table 1. Examples of lifestyles elements.**

| Main theorists | Component |
|---|---|
| **Simmel, Veblen, and Weber** | Diet, self-decoration (type of clothing and fashion), type of housing (decoration, architecture, and furniture), vehicle of transportation, ways of spending leisure and entertainment, mannerism (nobility, playfulness, smoking in public, number of employees, and their makeup) |
| **Adler** | All the behaviors, thoughts, and feelings of the person and his movement towards the goals |
| **Kluckhohn** | Personal consumption behaviors reflecting individual preferences, such as how to use cultural, recreational, and sports industries, how to play, and how to dress up |
| **Warner** | How to spend leisure time (recreation and sports tastes), religious differences, political values, family life patterns, marital relationships, and parenting styles |
| **Bourdieu** | Properties (luxury or cultural goods, such as houses, villas, yachts, cars, furniture, paintings, books, soft drinks, cigarettes, perfumes, and clothes), activities differentiating a person (such as sports, games and entertainment, and clothing), taking care of body, how to use language, and budgeting |
| **Veal** | Consumption, values, attitudes, demographic issues, gender differences, economic status, occupation, social classification, and participation in leisure activities |
| **Vyncke** | Values, life vision, aesthetic styles, and media preferences |

The exchange of large sums of money certainly significantly affects people's consumption patterns, including athletes and lifestyles. In reviewing the literature on sports sociology in this field, one can find studies related to consumer behavior in sports according to social, economic, and demographic levels. However, the target group in most of these studies includes non-athletes [11, 14, 16–19]. In another set of studies, factors related to athletes' lifestyles, such as addiction [20], exercise addiction [21], leisure and dropout [22], health management in life [23], health challenges, and gender differences [24] have been emphasized. As Kahma (2012) believes, in lifestyle studies and theories, sports subjects are often marginalized and ignored [15]. Roderick (2006) believed that less research has focused on the real-life of elite athletes in all its positive and negative aspects. Single-minded and obsessed with reaching the goal are among the unique characteristics of elite athletes that can also influence their lifestyle [25]. Professional athletes often experience independent living from their families due to immigration to other cities or countries. This independent life usually occurs at a very young age, posing many challenges for athletes and their families. Some team members, such as coaches and club managers, may significantly interfere with athletes' lifestyles, including their leisure time if their performance is impaired [26]. The above studies are extensive, and they can help understand the elite athletes' lifestyle, but none of them present all possible indicators of an athlete's lifestyle. Besides the lifestyle indicators, lifestyle shaping variables are also critical. In simpler words, the study of addiction, exercise addiction, leisure and dropout, health management in life, health challenges, and gender differences as factors related to athletes' lifestyles is a good action that can help to direct athletes to success. However, if we do not know the reasons for choosing certain lifestyles and athletes' tendency to specific actions and behaviors, our knowledge about the lifestyle of athletes will be superficial, a point that we are going to address in this study.

It can be concluded that lifestyle studies of elite athletes have often been conducted indirectly in three parts: leisure, consumption behaviors, and health-oriented lifestyle. Although they can help understand the lifestyle of elite athletes, it is believed that the lifestyle of elite athletes has a broader area. In the present era, lifestyles are faced with issues of multiplicity and diversity on the one hand and high speed of change on the other hand so that, even these changes in the lifestyle of society are a cause of curiosity and sometimes a concern of experts and socio-cultural policy-makers [27]. Therefore, if the lifestyle of different people in society is not desirable and there is no balance between different aspects of their lives, it can lead to critical problems and challenges for communities. We all know that the nature of sports is constantly changing from leisure and recreational activity in the past to a broader area with the concept of the sports industry. As one of the leading players in this industry, athletes are influenced by social, economic, political, and cultural factors. Athletes in this industry are not only consumers but also producers of many attractive competitions. According to the theories studied in this field (Veblen, Bourdieu, etc.), it can be concluded that these theories are often based on consumption or leisure. However, for analyzing elite athletes' lifestyles, a theory is needed to consider all aspects of their lives. This should be a theory that considers the athletic characteristics of athletes and their most important actions in sports and non-sports fields (on and off the field). A comprehensive study of athletes' lifestyles can be vital as they are role models and are inherently essential and influence different groups in society, especially young people. For example, 89.4% of Japanese adolescents identified role models for themselves in a study. In this case, sports players were the most famous role models [28]. William C. Cockerham believes all indicators show that research in the lifestyle field is at the center of sociological research in the 21st century [29]. As a result, theorizing about elite athletes' lifestyles is in line with the evolution of general theories of lifestyle will be. Although theories such as Adler, Simmel, Veblen, Bourdieu in the lifestyle area are beneficial, these theories are in general sociology.

Furthermore, we must have specialized theories in the sports sociology area about athletes' lifestyles. Presenting such theories can help better understand elite athletes' lives because general issues (such as what is seen in Table 1) cannot cover all the topics in the lives of elite athletes. For example, Dr. Kevin Fleming believes "Professional athletes are a unique breed. They attract millions of followers and harbor athletic skills and expertise that most only dream of having. From mainstream professional leagues like football, basketball, hockey, and baseball to elite tennis players, Olympians, and so on, they spend their lives pursuing childhood dreams that beat enormous odds. Many of them end up making millions and millions of dollars, but somehow, happiness and peace of mind are fleeting for them. So often it seems that these are the individuals caught up in drug abuse, alcoholism, anger issues, crime, and the list goes on ad nauseam" [30]. It means the handling of elite athletes' lifestyle needs a better and specialized theory which none of the general theories in the lifestyle area have this potential. Due to extensive scopes of athletes' lifestyles, previous studies have often focused on a few aspects. A comprehensive theory has not been provided about it in sports sociology. Finally, given that any policy and decision-making in social management requires the discovery of countless perspectives on the lifestyle of the people of society [31], a gap in research about elite athletes' lifestyles is evident in this situation. So, it is attempted to theorize elite athletes' lifestyles in this study.

## Materials and methods

Every researcher has his/her view of what constitutes truth and knowledge. These views guide our thinking, beliefs, and assumptions about society and ourselves, and they frame how we view the world around us, which is what social scientists call a paradigm [32]. In scientific research, the paradigm comprises perception, beliefs, and awareness about the different theories and practices employed in the study process [33]. Philosophical assumptions about the nature of social reality (known as ontology), ways of knowing (known as epistemology), and ethics and value systems (known as axiology) make up a paradigm [34]. The paradigm of scientific research, in turn, consists of ontology, epistemology, and methodology [33]. The basic beliefs, assumptions, and postulates of a paradigm are learned via the processes of socialization, telling researchers what is necessary, legitimate, and reasonable to study [35]. A paradigm thus leads us to ask specific questions and use appropriate approaches to inquiry. Several reasons led us to follow the interpretivism paradigm assumptions in this study. In interpretivism research philosophy, the researcher states that it is not easy to understand the social world based on principles. Interpretivist research philosophy says that the social world can be interpreted subjectively. The most outstanding attention here is given to understanding the ways through which people experience the social world. The interpretive research philosophy is based on the principle that the researcher performs a specific role in observing the social world. Using this philosophy, research is based and determined by the researcher's interests [33]. On the question of what reality is (ontology), the interpretivists believe that it is socially constructed and that there are as many intangible realities as people construct them. Reality is, therefore, mind-dependent and a personal or social construct. About epistemology, they believe that knowledge is subjective because it is socially constructed and mind-dependent. The truth lies within the human experience. Therefore, statements on what is true or false are culture-bound, historically, and context-dependent, although some may be universal. Within this context, communities' stories, belief systems, and claims of spiritual and earth connections find space as legitimate knowledge. From a methodological perspective, interpretative research aims to understand people's experiences. The research takes place in a natural setting where the participants make their living. The purpose of the study expresses the assumptions of the interpretative researcher in attempting to understand human experiences. Assumptions about

the multiplicity of realities also inform the research process [36]. Herein, we believe that the reality of Iranian athletes' lifestyle is local and subjective because we do not have a single perception of it, and every person has their definition. Also, the reality of lifestyle depends on personal or social conditions. Every person experiences their lifestyle, and true or false (or good or bad) of it depends on the social context. It means that the reality of lifestyle is influenced by diverse values such as personal, indigenous, religious, and national values. Furthermore, we should try to understand athletes' experiences in the natural settings of their lives to gain knowledge. Mackenzie and Knipe (2006) argue that the paradigm and research question determine which method will be most appropriate for gathering and analyzing data (qualitative/quantitative or mixed) [37]. Particular paradigms may be associated with specific methodologies. For example, a positivistic paradigm typically assumes a quantitative methodology, while a constructivist or interpretative paradigm usually utilizes a qualitative methodology [36]. Common designs in the interpretivism paradigm include ethnography, phenomenology, biography, case study, and grounded theory, which belong to the category of qualitative research [38].

In this study, a qualitative approach was used to understand elite Iranian athletes' lifestyles and develop a lifestyle typology. In the first phase of the study, Glaserian grounded theory was employed for several reasons. Despite the existence of theories, such as Veblen's leisure class theory or Bourdieu's theory, these theories do not seem to be appropriate and complete for analyzing the lifestyle of elite athletes. Because of this, it was necessary to theorize about elite athletes' lifestyles, and therefore, the selection of GT is justified. Glaserian and Straussian GT are the two main schools of GT [39]. In Glaser's opinion (1992), Strauss and Corbin's method is not related to GT, but it is a method founded on the "forced conceptual description." In this way, Glaser (1992) has used the term "Emergence vs. Forcing" to describe the difference between his method and Strauss and Corbin's method. Glaser (1992) believed that the axial coding method (Strauss and Corbin's method) is suitable for novice researchers due to its ease of implementation [40].

Glaserian GT also has other traits that can be useful in studying elite athletes' lifestyles. Thus, it was tried to provide a theory about elite athletes' lifestyles, but our theory must be based on reality, not researchers' ideals or other considerations. In this regard, Glaser (1992) mentioned that: "The theory must respect and reveal the perspective of the subjects and not that of the researcher." Since lifestyle is a social topic and people themselves determine their lifestyles, the descriptive theory must also incorporate the perspectives and opinions of the subjects. Thus, a theory that rises from the ground using the Glaserian GT method can be established. Another characteristic of Glaserian GT supporting its use in studying elite athletes' lifestyles is the basic social process (BSP). About BSP, Glaser and Holton (2005) said that: "Similar to all the GTs, generation of a BSP theory occurs around a core category. While, a core category is always present in a grounded research study, a BSP may not be so" [41]. The BSP is a category with two or more emergent phases resolving the group's main concern under study. It refers to the process of moving through a situation [42]. BSPs are labeled by a "gerund" ("Ing"). The concepts like cultivating, defaulting, centering, highlighting, and becoming are BSPs showing feeling in the process, change, and movement over time [41]. The following sections will discuss why and how the BSP appears in the study of elite athletes' lifestyles. However, regarding the discovery of the major concerns of the participants about elite athletes' lifestyles, it is clear that a process is required to address the concerns and improve the situation.

## Data collection and analysis procedures

Theoretical sampling was used to provide a theory about elite athletes' lifestyles in two phases. In Glaser and Strausss' (1967) and Glasers' (1978) approaches, theoretical sampling refers to

collecting, encoding, and analyzing data simultaneously to form a theory. Additionally, the researcher decides what data to collect and where to find them to improve the theory [43]. In the first phase, 19 experts aware of the Iranian athletes' lifestyles were recruited. These participants included coaches, educated athletes, sports managers, sports psychologists, sports sociologists who met specific requirements, such as authoring lifestyle papers, having solid relationships with athletes, and being introduced by the research team (the first interviewees). According to Glaser (1978), "generation of theory occurs around a core category" and indicates the main theme of a substantive area of inquiry. The core category is determined through iterative coding, memoing, theoretical sampling, and the theoretical sorting process. There are overlaps in these phases; they fluctuate back and forth and are not as clear-cut as they could be. For the second phase, interviews were continued with the participation of 44 Iranian national team elite athletes to establish a typology of athletes' lifestyles. They were asked to comment freely on various aspects of their lives and daily activities. They also talked about their preferences and interests. They were assured that their statements were confidential and would only be used for research purposes. The data of this phase were analyzed by reflexive thematic analysis method and considering the comprehensive model of the elite athletes' lifestyle of the first phase. As a result, dominant lifestyles were identified among the Iranian elite athletes.

Open coding, memoing, selective coding, and theoretical coding were used to theorize about elite athletes' lifestyles. Open coding is the first step towards discovering categories and their properties [40, 43, 44]. Case-based memos and conceptual memos were two types of memos written during this study's interviews and analysis. A case-based memo captured what the interviewer learned from the interview [45]. Conceptual memos or theoretical memos are other types of memos that help develop the theory. According to Glaser (1978), memos are a key part of the process, and without using them to write up an idea, the researcher is not conducting GT. Memos are building blocks of theory development in GT [46].

Selective coding means coding in terms of a core category. In selective coding, theoretical memos focus on aspects of the core category, and those aspects guide theoretical sampling. In this way, the theory can be enriched, and even the participants' main concern and how they resolve it can be revealed through the data by focusing on the main category. Theoretical codes "conceptualize how substantive codes may relate to each other as hypotheses to be integrated into a theory. They are emergent like the substantive codes; they weave the fractured story back together again. . . Theoretical codes give integrative scope, broad pictures, and a new perspective" [43]. Data collection and analysis were stopped when theoretical saturation was achieved. Theoretical saturation in qualitative research often means that what is said in the interviews is repetitive, and nothing new is received from the participants. Lastly, the relationships between the concepts were mapped out using conceptual maps. The conceptual map represents key concepts of the theory and how they relate to one another. For creating a practical conceptual map, the researcher must continue theoretical coding and have a clear understanding of the data and their relation to each other. Thus, a theory was developed with a coherent set of concepts that relate to each other, and all the data collected were adequately explained. The developed theory was presented to sports experts and athletes, and it was accepted and resonated with these audiences.

An explanation of the process was provided by which a GT was derived from the data in this section. Of course, we cannot describe the long process of emergence of all concepts and construction of theory. The information on how data analysis, theoretical memoing, and theoretical sampling techniques were used to develop a GT progressively was demonstrated in the following. In coding, it was tried to use the natural language of the participants. However, we

did not limit ourselves to in-vivo codes, and if better codes were inferred from participants' quotations, they were selected (see Table 2).

## Standards of research quality

Research quality standards are still a challenging debate among qualitative research specialists in sports science and health. Some methods, such as member checking and Inter-rater Reliability, have been widely used in sports to determine trustworthiness and rigor in qualitative research [47, 48]. In the present study, member checking was performed by summarizing and retelling data for the interviewees and then asking them to confirm their accuracy. Inter-rater reliability was also attempted using several researchers (multiple coders) to analyze interview data. However, Smith and McGannon (2018) criticize these methods of validating qualitative research while noting: "...Reliability does not make sense when collecting qualitative data." They argue that the chance of agreement is more often due to shallow and delicate interpretations, which opposes the goal of qualitative research in providing interpretive and rich insights. Therefore, in the present study, we tried to use more methods for rigor in qualitative research, such as some parallel position methods. Dependability is one of these criteria parallel to the conventional reliability criterion [35].

For a qualitative study to be considered reliable, it must be consistent and accurate. It is the responsibility of researchers to demonstrate this by providing an audit trail in which they provide a detailed description of their research trajectory and decision-making processes [35]. In the present study, all stages and procedures of the audit were tried to be described in detail and clearly. Data analysis examples in Table 2 and codes for each participant in Table 3 were provided to clarify these.

## Ethics approval and ethical issues

Ethical considerations are one of the necessary conditions for conducting research; in this study, the following instructions were done to observe these considerations:

1. The Ethics Board approved the proposal for this study of the University of Tehran.

2. A list of possible actions and strategies was provided in selecting interviewees and conducting interviews in accordance with Charmazs' titles.

3. Participants took part in the interviews voluntarily, and they had the right to withdraw at any stage, and their information was strictly confidential. Confidentiality was maintained by assigning a unique code number for each participant, and the number was used only to distinguish between participants' interviews.

4. All statements were anonymous prior to analysis, and the researchers were cautious not to disclose potential details, such as location, team, or names of participants.

5. According to ethical guidelines, all oral and written information (interview guide) was provided to the interviewees before the interview.

6. Study participants were all adults (age >18), and they gave their consent verbally. Reason for using verbal consent; It was the preservation of the identity of the participants. Because our participants were elite athletes, asking for written consent could create an atmosphere of mistrust. After voluntary attendance and verbal consent, all interviews were recorded, and the audio file of the interviews (in Persian) was available.

7. The interview environment was chosen with respect to the participant's consent so they could freely share their experiences.

**Table 2. Examples of open coding process.**

| |
|---|
| Q: What is lifestyle of elite athletes like? |
| *"An elite athlete's lifestyle goes beyond how they eat and dress up . . . We need to see what the athlete sees for his/her future. What are his/her goals regarding education? . . . Career development plans are also important, and the athlete must have a clear path to economic development".* |
| Open coding: |
| Eating habits |
| Dressing style |
| Educational vision |
| Occupational development plans |
| Economic development |
| *"Some athletes spend their leisure time with their families, that is, they do any activity with the family, if they want to watch a movie, they do it with the family, if they want to go shopping, they will go with family members" . . . "Many people today spend a lot of leisure time in the virtual world, and elite athletes are no exception" . . . "Showing luxurious and sumptuous activities has become an important habit in athletes" . . . "Since, some athletes are quickly seen in the media and judged, they prefer to do everything covertly. It means, these athletes prefer to spend their leisure indoors, such as at private parties and friendly small gatherings".* |
| Open coding: |
| Familial leisure |
| Virtual leisure |
| Sumptuous activities |
| Covertly |
| Indoors |
| Private parties |
| Friendly small gatherings |
| Q: Is there a way to make the lifestyle of elite athletes more professional? |
| *"We can say that elite athletes do not live professional lives, but part of that depends on them personally. The solution to this problem lies at various levels . . . Sports organizations must be professional and managed by professional principle . . . Sports colleges and universities are required to be established to support from athletes' education. Supporting athletes' academic education should be a priority in these universities . . . Developing and presenting a role model athlete is crucial for young athletes".* |
| Open coding: |
| Professional sports clubs |
| Establishment of sports schools |
| Suitable national role model |
| Academic supportive programs |

## Results

### Typology of elite athletes' lifestyles

The second stage of the study was conducted after theorizing about elite athletes' lifestyles. The theory advanced our understanding of lifestyle-shaping structures of elite athletes, lifestyle indicators, and even professionalization of their lifestyles (see Table 3 and Fig 1). Indeed, some indicators that seem to be important in athletes' lifestyles from the point of view of sports experts were extracted. In addition to being the initial guide of the interview in the second stage, these indicators were able to give a clear structure to our theory and organize the data obtained in the second stage. It should not be forgotten that due to the breadth of the concept of lifestyle and the possibility of drowning the interview in marginal and irrelevant notions, it was challenging to focus on relevant data. These indicators were like a guide to the destination. At the same time, the two-step process, according to some criteria of research quality such as rich rigor and sincerity, can create an appropriate theoretical structure and prevent

**Table 3. Categories, concepts, and emerged codes by participants (Experts).**

| Category | Concepts | Final codes | Interviewees (Experts) |
|---|---|---|---|
| **The lifestyle-shaping structures** | Macro level | Economic structure | E4, E11, E14 |
| | | Socio-cultural structure | E17 |
| | | Political-legal structure | E5, E13, E19 |
| | | Religion | E1, E7 |
| | Meso level | Global styles | E12, E15 |
| | | Traditional elites' styles | E2, E6, E10 |
| | Micro level | Familial situation | E15 |
| | | Individual ideals | E3, E8, E17 |
| | | Childhood deficiencies | E18 |
| **Indicators of elite athletes' lifestyle** | Responsibility | Social responsibility | E6 |
| | | Sporting responsibility | E2, E9 |
| | | Sportsmanship | E8, E16 |
| | Financial literacy | Foresight financial | E15 |
| | | Being thrifty | E5 |
| | | Financial advisor user | E10, E13 |
| | Consumption | Distinctive consumers | E5, E14 |
| | | Need-driven consumers | E19 |
| | | Cultural consumers | E8 |
| | Leisure | Familial leisure | E12, E17 |
| | | Closed | E1, E11, E18 |
| | | Virtual leisure | E3 |
| | | Sumptuous | E10 |
| | Professional mindset | Professional relations | E4, E7 |
| | | Punctuality | E14 |
| | Life vision | Educational vision | E1, E10 |
| | | Occupational development plans | E3, E19 |
| | | Economic development | E1, E11 |
| | | Reputation | E7 |
| | Competencies | Self-management | E14, E17 |
| | | Self-care | E6, E13, E14 |
| | Religious behaviors | Religious duties | E4 |
| | | Religious relations | E14 |
| | Personal issues | Marriage style | E5, E18 |
| | | Dependency | E10, E16 |
| **Professionalizing of lifestyle (BSP)** | Macro level | Professional sports governing body | E15 |
| | | Sports colleges | E 15, E19 |
| | | Academic supportive programs | E2, E9 |
| | | National role model | E16 |
| | Meso level | Sports schools | E3 |
| | | Professional sports club | E15, E17 |
| | Micro level | Goal setting | E11 |
| | | Imitation of a role model | E15, E18 |

researchers' mental bias or possible deviations due to personal prejudices [49]. In the second phase, five lifestyles were found among elite athletes based on reflexive thematic analysis. The qualitative data were coded, and based on the frequency of the emerged themes, the most relevant titles were selected for each lifestyle.

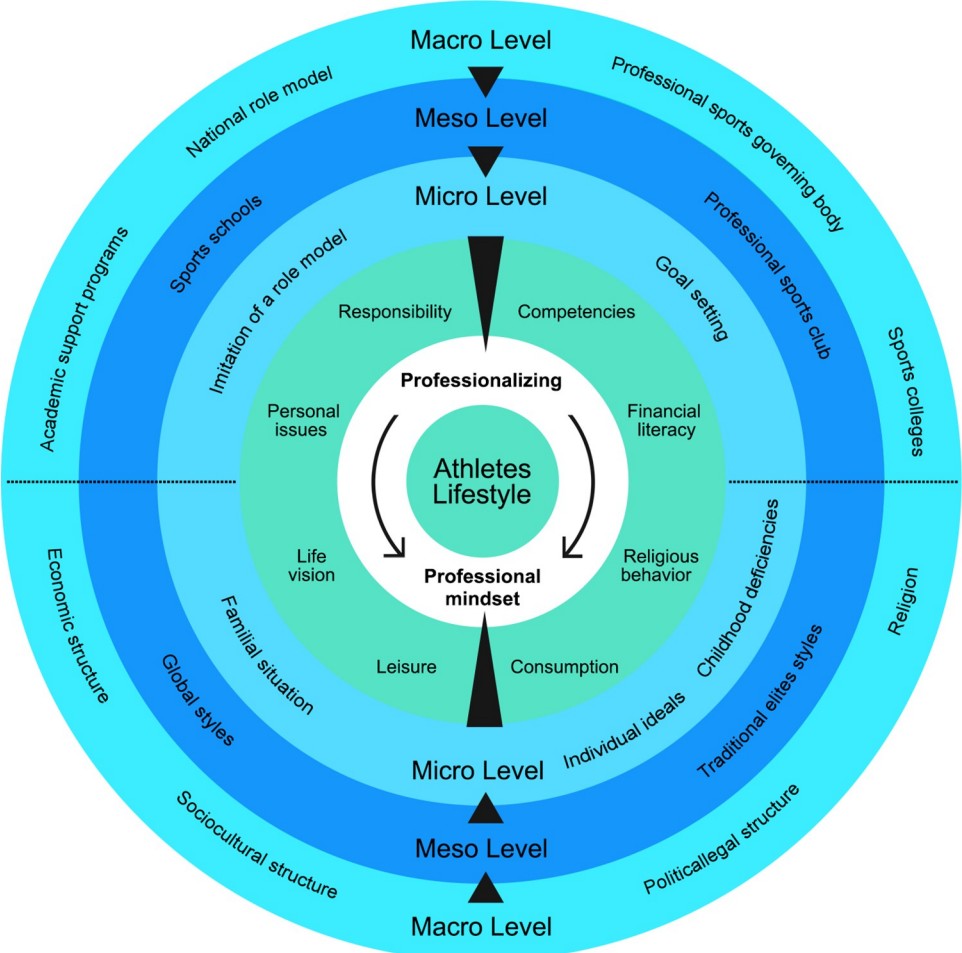

**Fig 1. Conceptual map of elite athlete's lifestyle.**

## Style 1: Consumerist style

Certain themes that indicated the importance of consumption for athletes were determined while analyzing the interviews. Athletes interested in consumerist lifestyles generated high numbers of codes about consumption, goods, new brands, cars, etc. Also, changing these things early was important to these athletes. These athletes tend to own and obtain products and goods beyond their basic needs. They had no special insight even when they were asked about important issues, like saving money, responsibility, professional mindset, etc. So, consumerism was chosen for their lifestyle. Consumerism is a style that represents negative aspects of modern life. Consumption is at the center of it. It means athletes of this lifestyle follow the consumerism culture. The importance of other aspects of life, such as professional thinking, competencies, life vision, financial literacy, and responsibility, is far less than material consumption for consumerist athletes (see Table 4).

## Style 2: Easygoing style

There were no notable highlights in the second set of the produced themes. Individuals in this group tended not to exhibit rapid changes in their life. Additionally, their decisions were usually immediate and based on the easiest option. When they were asked about life vision, goals,

**Table 4. Themes and emerged codes by participants (Athletes).**

| Themes | Codes | Interviewees (Athletes) |
|---|---|---|
| **Consumerist** | Shopping | A3, A7, A15, A18, A22, A25, A26, A33, A35, A40, A42 |
| | Change | A2, A4, A9, A15, A25, A38, A42 |
| | Car | A3, A8, A17, A20, A34, A40, A42 |
| | Stylish | A4, A8, A11, A35, A44 |
| | Gadgets | A12, A18, A26, A37 |
| | Luxury | A7, A20, A24, A34 |
| | New fashioned | A6, A7, A13, A34 |
| | Fashionable | A13, A25, A34, A35 |
| | Brands | A8, A13, A26, A35 |
| | Latest model | A17, A34 |
| | Travel | A4, A40 |
| | Differentiation | A11, A44 |
| | Clothes | A13, A35 |
| | Home-parties | A7, A15 |
| | Expensive | A35, A38 |
| | Newest | A7, A17 |
| | Appliances | A4, A24 |
| | Elegant | A44 |
| | Luxury- leisure | A20 |
| | Personal tailor | A25 |
| | Smart phone | A4 |
| | Make up | A11 |
| | Newer | A7 |
| | Charm | A15 |
| **Easygoing** | Unplanned | A5, A9, A19, A27, A36, A39 |
| | Relaxing | A5, A12, A19, A27, A30, A39 |
| | Passive leisure | A15, A17 |
| | Apathetic | A9, A20 |
| | Stability | A31, A39 |
| | Influenced-by-friends | A10, A23 |
| | Immediate-decision | A3, A16 |
| | Comfort | A5, A19 |
| | Care free | A28 |
| | Apathetic | A23 |
| | (Do) Nothing | A43 |
| | Indifferent | A27 |
| | Incurious | A31 |
| **Socially Useful** | Society | A1, A4, A11, A22, A26, A34, A42 |
| | Being-useful | A4, A14, A26, A31, A38, A40, A44 |
| | Good-athlete | A12, A26, A40 |
| | Role-model | A11, A22, A29 |
| | Social-responsibility | A1, A6, A32 |
| | Impact | A10, A29, A44 |
| | Social-expectation | A22, A42 |
| | Social-duty | A1, A13 |
| | Positive | A12, A39 |
| | Reputation | A22 |
| | Charity | A34 |
| | Social face | A4 |

*(Continued)*

**Table 4.** (Continued)

| Themes | Codes | Interviewees (Athletes) |
|---|---|---|
| **Profit-Oriented** | Profit | A6, A9, A16, A21, A30, A43 |
| | Money | A6, A11, A16, A18, A24, A43 |
| | Make money | A9, A19, A32, A36 |
| | Investment | A14, A21, A32, A36 |
| | Economic activities | A6, A13, A27, A43 |
| | Ownership | A15, A18 |
| | Cost management | A16, A34 |
| | Gaining | A12 |
| | Saving | A6 |
| | Financial advisor | A6 |
| | Cost-control | A16 |
| | Retirement | A9 |
| | Financial goals | A40 |
| | Cost reduction | A23 |
| | Budgeting | A9 |
| | Planning | A6 |
| **Professional** | Balanced | A8, A10, A14, A17, A23, A28, A29, A33, A41 |
| | Growth | A8, A28, A29, A33 |
| | Punctual | A6, A17, A23, A41 |
| | Self-discipline | A12, A14, A23 |
| | Public-relationship | A2, A41, A44 |
| | Public-behavior | A2, A22 |
| | Training principles | A2, A41 |
| | On time | A12, A10 |
| | Timely | A23 |
| | Accountable | A44 |
| | Sport agent | A28 |
| | Logical | A6 |
| | Consumption | A40 |
| | Privacy | A8 |
| | Goal | A33 |
| | Career development | A28 |
| | Academic development | A8 |
| | Sportsmanship | A22 |
| | Media | A41 |
| | Technical capability | A17 |
| | Prioritization | A8 |
| | Task | A14 |
| | Advisor | A23 |

plans, responsibility, financial matters, etc., it was found that these athletes had no clear answers and have postponed the decision about these issues for the future. This lifestyle type is called easygoing. Easygoing style is a style that represents incuriosity towards oneself, surrounding people, society, and life. These athletes do not have specific life plans. They fail in various aspects of life and may even be frustrated with refining and improving their life (see Table 4).

### Style 3: Socially useful style

The third set of emerging themes was about social concepts. In other words, in the themes produced by the athletes of the third group, social orientation was more prominent. In addition to social issues, themes related to consumption were also discussed in this group, but they were much less frequent than social issues. Athletes of this lifestyle were found to have a socially helpful lifestyle. A socially useful lifestyle demonstrating being a useful member of society is the most critical type of lifestyle. Socially useful athletes have a great deal of social interest and activity. They strive to perform their duties as part of society effectively. Foresight and life vision emphasized that reputation was more important than money for these athletes (see Table 4).

### Style 4: Profit-oriented style

The fourth group of athletes produced a more prominent theme of money and profit. This lifestyle type was called profit-oriented because money was necessary, but it was essential to maintain and increase it (making a profit). In addition to their regular consumption and leisure, these athletes were more oriented towards closed leisure, which means they were not interested in showing off their lives. Profit-oriented style is a style in which maintenance and development of financial resources are more important than sports activities. In other words, profit-oriented athletes manage all aspects of life to maximize their financial benefit in any situation. They avoid actions endangering their growing status and do not waste money for excessive consumption. In general, it can be said that profit-oriented athletes move towards the development of their financial resources with the lights off (secretly and without attracting attention) (see Table 4).

### Style 5: Professional style

There was a precise balance between life's different aspects in the fifth set of emergent themes, different from the previous four styles. In this group, athletes believed that every athlete should strive to strike the precise balance between factors, such as responsibility, professional mindset, consumption, life vision, competencies, financial issues, etc. In addition to personal effort, the key to balancing sports and non-sports aspects of life was receiving help from experts as a consultant. Professional is the name given to this lifestyle type. The professional lifestyle shows athletes looking for the best performance in sports according to scientific and ethical principles. Professional athletes seek to create the best balance among different aspects of life (see Table 4).

## Discussion

A qualitative approach was used here to theorize the lifestyle of elite Iranian athletes. Elite athletes' lifestyles can be defined as a framework influenced by individual choices and socioeconomic forces of society. This framework analyzes athletes' behaviors and actions related to sports and non-sporting aspects of their lives. Our finding has some aspects in common with theories proposed by Adler and Bourdieu.

Even though structures of society influence individual choices, they do not impose any particular lifestyle on the Iranian elite athletes. It means an elite Iranian athlete's lifestyle is formed through his/her choices. At the macro-level, economic, political, legal, socio-cultural, and religious factors were emphasized. Elite athletes' lifestyles are influenced by global styles and traditional elites' styles at the meso-level. Athletes construct their preferred lifestyle depending on all of the factors mentioned above and individual circumstances (familial situation, personal

ideals, and childhood deficiencies). Athletes' preferred lifestyles are selective. One can see a wide range of lifestyles despite similar macro and meso conditions and even similar individual conditions (based on the study's second phase).

In this context, Bourdieu uses the term habitus. The habitus is the central concept of Bourdieu's attempt to transcend sterile dualism between objectivism and subjectivism. Bourdieu rejects the idea that individuals' daily actions can be explained by individual decision-making or supra-individual determinative social structures [50]. Bourdieu (1977) used habitus to explain the agency-structure relationship in lifestyle dispositions [12]. Habitus can be viewed as a cognitive map or a set of perceptions routinely guiding and evaluating a person's choices and options. It provides enduring dispositions toward acting deemed appropriate by a person in particular social situations and settings. Habitus is created by exterior social structures and conditions and individual preferences, inclinations, and interpretations. The dispositions reflect both established normative patterns of social behavior and habitual and intuitive actions [51].

Although Bourdieu attaches great importance to social structures and external conditions in creating Habitus, Habitus also limits individual choice. According to Williams [50], the habitus is not a barrier to choosing. In other words, the habitus explains patterns of social behavior and habits of action, and the person makes a conscious decision about the action they will take. In this regard, Pang, Macdonald, and Hay [52], in a study entitled "Do I Have a Choice?" While using Bourdieu's concepts, such as habitus, emphasized the choice of lifestyle by Chinese youth [52]. As a result, one can say that the Iranian elite athletes choose their lifestyle or even create it despite macro-level structures and meso-level conditions.

Moreover, according to the participants, inappropriate lifestyles of the Iranian elite athletes are not the result of social structures and external conditions but are a consequence of the choices they make in life. The difference between Bourdieu's theory and ours is that Bourdieu's (1984) theory emphasized the importance of exterior social structures and conditions in individual decision-making. It means Bourdieu only uses the "social structures and conditions" term and does not specify precisely these structures [12]. Nevertheless, in our study, the structure and conditions are perfectly customized for the lives of athletes. In simple terms, Bourdieu's theory can be used to understand the lifestyle of elite athletes like other citizens. Nevertheless, in this case, the distinguishing factors will be general variables such as the division of day and night, the type of leisure, ways of socializing, furniture and home, and the etiquette of speaking and walking. Also, if we use Bourdieu's theory for understanding elite athletes' lifestyles, the exterior social structures and conditions, individual preferences, inclinations, and interpretations will remain anonymous. However, here, we identified the lifestyle-shaping structures at three levels that significantly have a sporty nature at the meso-level, such as global and traditional elites' styles. Bourdieu's theory has no information about this and could not guide sports scholars in learning elite athletes' lifestyles.

Are all the Iranian elite athletes the same lifestyle, or can there be differences between their lifestyles? This is another point to consider. It is possible to understand the existence of different lifestyles among athletes by reflecting on Adler's theories (1956), according to which lifestyle is viewed as an individual attribute and considers the number of human beings to be a different lifestyle. Petev [53] also confirmed that different lifestyles exist even within a social class. Using data from the past four decades in American society, Petev [53] pointed to changes in the diversity of lifestyles within and between different classes. Diversity within and between social classes means that in addition to the fact that lifestyles of different social classes are very different, even in today's changing conditions, changes in the lifestyle of people within a social class are noticeable [53]. Consequently, if we claim that there is only one class or stratum of the Iranian elite athletes, it does not make sense, and we can expect that there are different

lifestyles among the Iranian athletes. As in the second phase of our study, five types of lifestyles were identified among the Iranian elite athletes.

The term lifestyle reflects groups' distinctive style of life with a specific status (or distinction). Furthermore, it implies individuality, self-expression, and stylistic self-consciousness in contemporary consumers' culture. A person's body, clothes, speech, leisure pastimes, eating and drinking preferences, home, car, choice of holidays, etc., indicate individuality of taste and style of that person [5]. In general, the indicators mentioned above are still mostly used in studying lifestyle differences, which can be called consumption. As one of the classic sociologists, Thorstein Veblen (1899) refers to the lifestyle of the leisure class in his work as "the theory of leisure class." Veblen (1899) uses the term "conspicuous consumption" to describe the lifestyle of the leisure class. Conspicuous consumption refers to the use of goods regardless of their application, which are useless activities that contribute neither to the economy nor material production of the useful goods and services required for the functioning of society. This form of consumption brings reputation to the individual. Veblen (1899) pointed out that it does not just have power and wealth that matters, but these things need to be displayed to the public because only then wealth and power are respected [9]. As mentioned earlier, consumption indicators remain one of the most important lifestyle indicators since it seems that they have good capabilities in displaying distinction. Law, Bloyce, and Waddington (2020) performed a noteworthy study in this regard. Using the concept of conspicuous consumption, these scholars discussed the consumption behaviors of professional footballers in England. According to Law, Bloyce, and Waddington [10], the lifestyle of successful players involves the purchase and display of expensive luxury items, such as clothes made by famous designers, watches, and cars. They were also aware of their consumption behaviors and stated that they intended to display wealth and power through conspicuous consumption inside and outside the club. Finally, Law, Bloyce, and Waddington [10] inferred that pattern of conspicuous consumption in professional footballers might have some problematic aspects. For example, footballers are expected to adapt to the established patterns of conspicuous consumption within the club, and those who do not act in this way are likely to face various kinds of sanctions [10]. Therefore, it can be concluded that although consumption indicators are indeed useful for studying the lifestyle of athletes, if the lifestyle of elite athletes is analyzed only based on consumption, then the study will be superficial. For analyzing the lifestyle of elite athletes, more stable indicators are needed that are more relevant to the nature of sports activities. In this context, one can refer to two studies to support the need for new indicators. Cockerham [51], in a study entitled "Medical Sociology on the Move: New Directions in Theory," stated that a theory is required about healthy lifestyles to discuss it. Cockerham [51] noted that sociologists have neglected to recognize the effects of lifestyles on behavior and ultimately on health. According to Cockerham [51], the lack of research may be partly due to Veblen's theory and limitation of lifestyle to upper-class styles, equating it with conspicuous consumption. Then, Cockerham [51] cited from the study by Giddens (1991) as follows: "It was a major mistake to suppose that lifestyles are confined to those who are in more privileged materialistic circumstances. Everybody has a lifestyle, even the poorest of the poor". Cockerham [51] provides an updated theory of healthy lifestyles by reviewing various theories, especially Bourdieu's theory. The health lifestyles' paradigm is composed of different parts, such as class circumstances, age, gender, race/ethnicity, collectivities, living conditions, socialization experience, life choices, life chances (Structure), dispositions to act (Habitus), practices (Action), and health lifestyles (Reproduction). In the health lifestyles' theory, alcohol use, smoking, diet, and exercise are the most common practices (indicators). Physical check-ups and automobile seat belts were also added to the list of actions. Using the word "etc.", Cockerham (2013) established that health lifestyles' indicators go beyond the items mentioned above. Therefore, there will be the

possibility of developing healthy lifestyles' indicators in the future [51]. In another study, Vyncke (2002) emphasized the usefulness of lifestyle as a market segmentation tool. These studies are classified as "psychographic" studies, which use the AIO (activities, interests, and opinions) surveys to present lifestyle typologies. Vyncke (2002) introduced new approaches to constructing lifestyle typologies by incorporating more stable concepts, such as values, aesthetic styles, and life visions. Vyncke (2002) found that values, aesthetic styles, and life visions–either alone or in combination–can lead to very balanced and meaningful lifestyle typologies [54]. In conclusion, lifestyle indicators should be proportionate to the study area, as demonstrated by Cockerham [51] in health lifestyles' theory and Vyncke (2002) in market segmentation research. Additionally, lifestyle indicators seem to evolve with time.

Here, lifestyle indicators were identified for the Iranian elite athletes in another part of our study. It is believed that these indicators are very appropriate for analyzing the lifestyle of elite athletes. These indicators covered obvious issues, such as consumption, and included the most important actions of athletes concerning sports affairs in their lives. Elite athletes' lifestyles are characterized by responsibility, financial literacy, consumption, leisure, professional mindset, life vision, competencies, religious behaviors, and personal issues. Additionally, each indicator included different sub-modes. These indicators and their modes can determine different types of athletes' preferred lifestyles by showing the difference between them. To the best of our knowledge, no study mentioned all the above indicators for athletes' lifestyles. As a result, comparing similarities between our indicators and literature on athletes' lifestyles was challenging. Our study was qualitative, making comparisons of details of our findings with the literature largely unnecessary. However, some studies in this field are similar to part of our findings. According to Arai, Ko, and Ross (2014), one of athlete brand images' components includes "marketable lifestyle," meaning "an athlete's off-field marketable features." A marketable lifestyle has characteristics that reflect athletes' values and personalities. Three sub-dimensions of the marketable lifestyle include life story, role model, and relationship effort [55].

The life story of an athlete is an off-field story introducing a message and representing their values. Role model refers to an athlete's ethical behavior that society deems worthy of imitation. Relationship effort refers to an athlete's positive attitude toward interaction with fans, spectators, sponsors, and the media. Arai, Ko, and Ross (2014) inferred that achieving status depends on outstanding on-field performance and the distinctive lifestyle of the athlete. Because sports consumers can see athletes' lifestyles, researchers used lifestyles instead of athletes' personalities when creating the athletes' brand [55]. The similarity between our study and the above research can be discussed in two ways. First, there is a distinct feature of lifestyle whereby different types of lifestyles can be identified for athletes based on the proposed indicators.

Second, there is a similarity between the results of the study by Arai, Ko, and Ross (2014) and our study regarding "sportsmanship" in responsibility and "professional relationships" in a professional mindset. Sportsmanship is a multidimensional concept [56], but it is a moral issue in general. In this regard, William R. Reed said, "It [sportsmanship] is a word of exact and uncorrupted meaning in the English language, associated with an understandable and basic ethical norm." Moreover, Henry C. Link said, "Sportsmanship is probably the clearest and most popular expression of morals" [57]. Sportsmanship is not the topic of our discussion. However, it is clear that sportsmanship is comparable to role models, "An athlete's ethical behavior that has been determined to be worth emulating by society."

In the same way, "professional relationships" can also be compared with the concept of "relationship effort," which refers to a positive attitude toward interaction with fans, spectators, sponsors, and the media. It is believed that our study indicators are appropriate for analyzing the lifestyle of elite athletes. Lifestyle should not only be defined by the accepted items like consumption and leisure. According to Horley, Carroll, and Little (1988), "an acceptable definition

of lifestyle should include broadly-defined purposive behavior, which expresses underlying values and attitudes." Hence, lifestyle is viewed as a distinct behavior representing a characteristic pattern of values and attitudes [2]. Although values and attitudes are a wide area to study, athletes' attitudes towards items, such as responsibility, financial literacy, professional mindset, life vision, etc., are significant while analyzing athletes' lifestyles. For example, Tainsky and Babiak (2011) stated that sports and philanthropy are becoming entwined, and athletes gain more attention outside of their sports roles. Their study focused on socially responsible behavior among athletes, which showed the differences between them [58]. Therefore, concepts like responsibility can provide information about an elite athlete's lifestyle because athletes behave differently in this regard.

Another facet of our results was the typology of elite athletes' lifestyles. Adler (1956) stated that lifestyle is an individual attribute, and there are as many types of lifestyles as there are people. However, providing a lifestyle typology for elite athletes is more valuable, as similar measures (e.g., correctional and educational) can be prescribed for different lifestyles. Five types of lifestyles were identified among elite athletes. This typology was based on thematic analysis and was limited to 44 athletes who participated in the interview. Among the participants, 14 athletes had a consumerist lifestyle, four had an easygoing lifestyle, five had a socially useful lifestyle, 13 had a profit-oriented lifestyle, and eight had a professional lifestyle. The features of each style were discussed in the previous section. Also, the number of athletes in each style can provide useful tips. More than 30% of the studied Iranian elite athletes lived with a consumerist lifestyle. In addition, according to themes of this style, Veblen's leisure class theory best describes the lifestyle of this group of athletes. The prevalence of conspicuous consumption among Iranian athletes illustrates its universality, as discussed in other studies, such as Law, Bloyce, and Waddington's [10] study conducted in England. Conspicuous consumption may have variant causes and consequences in different countries, but it is not the topic of this paper. Approximately 10% (4 athletes) of the athletes had an easygoing lifestyle. There were only four athletes with easygoing lifestyles in this study, but that number might be significant compared to a larger population. Also, based on the themes that emerged in this style, the easygoing lifestyle is not admirable. Therefore, it is necessary to consider educational and correctional counseling for this lifestyle. However, further studies are recommended before deciding in this regard. Around 12% of athletes had a socially useful lifestyle. Community-oriented nature of this lifestyle makes it so appealing to society, and it seems that even athletes with this lifestyle are appreciated by society. However, judgments can be relative in this regard. There is no doubt that social affairs can help athletes build a positive image. The term "socially useful lifestyle" was also used by Adler to describe people with high social interest. A significant percentage of the interviewed athletes, about 30% (13 athletes), had a profit-oriented lifestyle. These athletes placed the most significant importance on preserving and developing financial resources. No discussion is needed here about whether this lifestyle is appropriate or not. It appears that a short period of participation in professional sports on the one hand and many problems faced by the retired athletes, especially financial problems, on the other hand, are sufficient reasons to justify this lifestyle. Athletes with a professional lifestyle accounted for 18% (8 people) of the interviewed athletes. Balancing various aspects is the essential characteristic of this lifestyle. Hence, the professional lifestyle here does not refer to the athlete who receives money for his/her sporting activities.

As a general conclusion, based on the five types of lifestyles identified, elite athletes should be guided to move from a consumerist and easygoing lifestyle to another. Professional and socially useful lifestyles seem to be the most desirable and beneficial for elite athletes. Any judgment regarding whether existing lifestyles are desirable or undesirable will require further research.

## Conclusions

Lifestyle is a popular concept that is easy to understand but is scientifically challenging. Elite athletes have received a great deal of attention in society in recent decades due to the rapid development of the sports industry. Therefore, the following question is raised: What is an athlete's life like? Or, to put it more scientifically, what is their lifestyle? This question was raised in this study since; citizens of the community find it important. However, there was no comprehensive theory about the lifestyle of elite athletes. In analyzing the lifestyle of elite athletes, restricting it to matters such as consumption and fashion would lead to a superficial interpretation. The Glaserian GT method enabled the researchers to develop a theory about elite athletes' lifestyles. Compared to the Strauss and Corbins' method or systematic approach of GT, this method has some critical differences. Attention and focus on participants' concerns, staying in the research environment for a long time to discover participants' main concerns, coding flexibility, and the emergence of a BSP around the main category were among some of the differences. The Glaserian GT approach provided a useful tool for understanding elite athletes' lifestyles. The researchers analyzed elite athletes' lifestyles from different perspectives using this approach. In addition to identifying lifestyle components of elite athletes, the major concerns of participants were presented, along with how to address them over time. According to Glaser (1978, 1996), any emergent theory should account for patterns and processes of behavior grounded in the data [43, 59]. These processes occur over time as stages, and as they change, they will come in a sequenced manner with respect to each other. In this study, the BSP was named as "professionalizing" primarily because the main concern of the participants and the core category were "lack of professionalism in life" and "professional mindset," respectively. Professionalizing as a process describes stages of resolving the athletes' lifestyle problems and becoming professional athletes. However, the fact is that becoming a professional athlete has not been well understood. Professionalism in sports does not only mean applying sports skills but also applying them to the other aspects of an athlete's life. This theory is still very young, and more qualitative studies on elite athletes' lifestyles are needed to refine it.

Moreover, the qualitative method was used in this study to identify dominant lifestyles of the Iranian elite athletes, including consumerist, easygoing, socially useful, profit-oriented, and professional lifestyles. Thus, carrying out quantitative research based on indicators of this research in a larger population to confirm the five types of lifestyles identified here. In general, our findings can be considered a specific theory for the lifestyle of elite athletes (regardless of the country of research). Although other specific variables may be added to each section of the theory in other countries, we now know the lifestyle indicators of elite athletes that can be validated in research in other societies. The present study's findings can also be used by coaches, sports psychologists, and sports managers to manage elite athletes better. Also, based on the findings, several training courses can be provided for elite athletes, especially at a young age, to choose more balanced lifestyles.

## Supporting information

**S1 File. Excerpts of the transcripts.**
(DOCX)

## Acknowledgments

The author would like to thank all the participants for their generous contribution of time and insight.

## Author Contributions

**Conceptualization:** Ehsan Mohamadi Turkmani, Hamid Reza Safari Jafarloo.

**Data curation:** Ehsan Mohamadi Turkmani.

**Formal analysis:** Ehsan Mohamadi Turkmani.

**Investigation:** Ehsan Mohamadi Turkmani.

**Methodology:** Ehsan Mohamadi Turkmani, Hamid Reza Safari Jafarloo.

**Project administration:** Ehsan Mohamadi Turkmani.

**Software:** Ehsan Mohamadi Turkmani.

**Supervision:** Ehsan Mohamadi Turkmani.

**Validation:** Ehsan Mohamadi Turkmani, Amin Dehghan Ghahfarokhi.

**Visualization:** Ehsan Mohamadi Turkmani.

**Writing – original draft:** Ehsan Mohamadi Turkmani.

**Writing – review & editing:** Ehsan Mohamadi Turkmani, Hamid Reza Safari Jafarloo, Amin Dehghan Ghahfarokhi.

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
