## [Decision Letter · Decision Letter 0]

4 Feb 2022

PONE-D-21-38970Elite Athletes' Lifestyles: Consumerism to ProfessionalismPLOS ONE

Dear Dr. Ehsan Mohamadi Turkmani,

Thank you for submitting your manuscript to PLOS ONE. After careful consideration, we feel that it has merit but does not fully meet PLOS ONE’s publication criteria as it currently stands. Therefore, we invite you to submit a revised version of the manuscript that addresses the points raised during the review process.

We look forward to receiving your revised manuscript.

Kind regards,

Rogis Baker, Ph.D

Academic Editor

PLOS ONE

Journal Requirements:

3. Please remove all personal information, ensure that the data shared are in accordance with participant consent, and re-upload a fully anonymized data set. 

Reviewers' comments:

Reviewer's Responses to Questions

**Comments to the Author**

1. Is the manuscript technically sound, and do the data support the conclusions?

Reviewer #1: Yes

Reviewer #2: Partly

2. Has the statistical analysis been performed appropriately and rigorously? 

Reviewer #1: Yes

Reviewer #2: Yes

3. Have the authors made all data underlying the findings in their manuscript fully available?

Reviewer #1: Yes

Reviewer #2: Yes

4. Is the manuscript presented in an intelligible fashion and written in standard English?

Reviewer #1: Yes

Reviewer #2: Yes

5. Review Comments to the Author

Reviewer #1: Thank you for your invitation to this research.

First of all, congratulations to the authors. The research idea and design is nice. The research design looks good.

Introduction, method, statistical analysis and discussion of the findings in the research were done very well. The results of the research provide an innovative perspective by contributing to the literature.

I think that the quality and content of the pictures in the text should be improved.

Reviewer #2: Thank you to the author(s) for submitting their manuscript to the Plos One Journal. Please see the page-by-page comments below. There seemed to be a couple of main issues that, if addressed, I think would enhance the research. For one, I question the need and purpose of the paper in relation to theoretical and practical implications of the research. That is, why there is a need to study lifestyles, what problem does it help with better understanding, and how the knowledge will be used and by who. Additionally, I believe the theoretical rationale for why the reviewed theories were not used, can be strengthened. I hope the author(s) are able to reflect on the comments and feedback in their future iterations of their paper.

Page 6: Paragraph above table 1: Could the author(s) also unfold concisely the studies that have used athletes and what to further explore in this research that the current paper/research addresses?

Page 7: It is believed paragraph: In another set of studies sentence and the remaining components of this paragraph: See previous comment as it seems here the athlete studies are being addressed. Could the author(s)articulate and strengthen the differentiation or distinction between the points in the paragraph above the table and the research included below?

Page 8: So far paragraph: Therefore, if lifestyle: The research prior could be better organized and tighter around these points. As such, was the health orientated lifestyle addressed.

Page 8: I am still unclear on how the reviewed theories could not be used to comprehensively explore the different aspects of their life—on and off the field. I think this rationale can be strengthened. As a suggestion, the author(s) might add a bit more of a critique in the beginning aspects of the paper where the theories are reviewed.

Page 9-11: Methods: could the author(s) highlight their epistemological and ontological assumptions/paradigmatic of their research.

Page 13: The components of the findings here might be better served as data collection and analysis procedures sections within the methods. Further, an added component on quality research standards would be appropriate.

Smith, B., & McGannon, K. R. (2018). Developing rigor in qualitative research: Problems and opportunities within sport and exercise psychology. International Review of Sport and Exercise Psychology, 11, 101-212.

Smith, B., Sparkes, A., & Caddick, N. (2014). Judging qualitative research. In L. Nelson, R. Groom, & P. Potrac (Eds.), Research methods in sports coaching (pp. 192–202). Routledge.

Sparkes, A. C., & Smith, B. (2014). Qualitative research methods in sport, exercise and health: From process to product. Routledge.

Page 14: There is a need to organize and describe these codes across the participants. Descriptions in what their deeper meaning are into a coherent description of the groups of open coding would be useful. See previous comment as it may open space for this.

Page 15: How were the codes/groupings in phase one used or informs phases 2?

Page 17: Style 2: Individuals in this group: Individuals in this group seem to lend itself to the idea that only part of athletes discussed this component. But, couldn’t there be integration across these themes/styles that could be involved in different aspects of the life, for different things, etc. Please clarify.

Page 19: Discussion: Some of the key points seem to run together in the discussion. Further, it would be useful to compare and contrast the theory generated from the research and Bourdieu. Some of the discussion involves articulating Bourdieu’s theory in relation to the findings, without as much consideration for the theory generated from the research. Based on this, I again, question, why couldn’t Bourdieu have been used in this research as I had indicated in a couple of other comments. Nonetheless, it would be useful to critique and compare and contrast Bourdieu to the findings. A final recommendation would be to include practical implications of the research findings. That is, who does the knowledge from this research address, what problem does it help solve or gain a better understanding towards, and how does this knowledge get used in practical application. Perhaps, there could be some indication of this in the introduction section from which the discussion would tie together in response the new knowledge generated from the findings.

6. PLOS authors have the option to publish the peer review history of their article (what does this mean?). If published, this will include your full peer review and any attached files.

Reviewer #1: **Yes: **Zeki AKYILDIZ

Reviewer #2: No

---

## [Author Response · Author response to Decision Letter 0]

13 Apr 2022

Dear Editor-in-chief/Reviewers 

We want to express our gratitude for creating the opportunity to improve the quality of our manuscript. It was very gratifying for us to get the favorable opinion of the reviewers in the first stage. We also appreciate the detailed comments of the reviewers, and we believe that these comments and possible future comments can significantly improve the quality of our manuscript. We highlighted the changes in the manuscript with colored text as follows. 

1) In response to the first reviewer's comment about the quality and content of the pictures in the text, we redesigned the main model of the paper in Photoshop in compliance with PLOS ONE's requirements. 

2) We explained more in the introduction why the reviewed theories were ineffective for elite athletes. Additionally, we highlighted the differences between the above and below studies of table 1. 

3) In the method section, we described our paradigmatic assumptions. Here, we used the valuable sources recommended by the reviewer.

4) We added a section on research quality standards in the method section and separated data collection and analysis procedures. 

5) To better organize the codes across participants, a column has been added to table 3, and table 4 has been added. Tables 3 and 4 show the codes generated by each participant. 

6) For the section on Typology of Elite Athletes' Lifestyles, we have added an explanation about using the first phase codes in phase 2.

7) Concerning style 2 (easygoing style), we can say that easygoing athletes may be placed in other styles since they make immediate decisions and do what is easy for them. One day they might be a consumerist, and another day they might prefer social activities. As a result, they were assigned a particular style (easygoing) because they lacked the necessary stability in other styles.

8) We have further explained our study's differences from Bourdieu's theory in the Discussion section. Additionally, we discussed the possible applications of our findings. Of course, we must note that our findings were based on a qualitative study, and the possibility of generalizing the results is weak.

We hope to satisfy the editor-in-chief and the reviewers, although the authors would be happy to announce their readiness for possible future revision.

Best wishes 

Research team

---

## [Editor Report · Decision Letter 1]

18 May 2022

Elite Athletes' Lifestyles: Consumerism to Professionalism

PONE-D-21-38970R1

Dear Dr. Ehsan Mohamadi Turkmani,

We’re pleased to inform you that your manuscript has been judged scientifically suitable for publication and will be formally accepted for publication once it meets all outstanding technical requirements.

Kind regards,

Rogis Baker, Ph.D

Academic Editor

PLOS ONE
---

## [Editor Report · Acceptance letter]

8 Sep 2022

PONE-D-21-38970R1 

Elite Athletes' Lifestyles: Consumerism to Professionalism 

Dear Dr. Mohamadi Turkmani:

I'm pleased to inform you that your manuscript has been deemed suitable for publication in PLOS ONE. Congratulations! Your manuscript is now with our production department. 

Kind regards, 

on behalf of

Dr. Rogis Baker 

Academic Editor

PLOS ONE